# Human milk-based fortifier is associated with less alteration of milk fat globule size than cow milk-based fortifier

Yurika Yoshida[1☯], Minami Azuma[1☯], Haruhiro Kuwabara[1], Tokuo Miyazawa[1], Yuya Nakano[1], Kazuna Furukawa[1], Keli M. Hawthorne[2], Masahiko Izumizaki[3], Takashi Takaki[4], Mari Sakaue[5], Katsumi Mizuno[1,6]*

1 Department of Pediatrics, Showa University School of Medicine, Tokyo, Japan, 2 Department of Pediatrics, Dell Medical School, Dell Pediatric Research Institute, University of Texas at Austin, Austin, Texas, 3 Department of Physiology, Showa University School of Medicine, Tokyo, Japan, 4 Division of Electron Microscopy, Showa University School of Medicine, Tokyo, Japan, 5 Analysis Systems Solution Development Department, Hitachi High-Tech Corporation, Tokyo, Japan, 6 Japanese Human Milk Bank Association, Tokyo, Japan

☯ These authors contributed equally to this work.
* katsuorobi@med.showa-u.ac.jp

**Data Availability Statement:** All relevant data are within the manuscript and its Supporting information files.

## Abstract

We aimed to evaluate if human milk-based fortifier (HMBF) affects human milk fat globule (MFG) size less than cow milk-based fortifier (CMBF), which may impact overall infant feeding tolerance. Measurements of donated human milk were performed before fortification as well as at 1 hour, 24 hours, and 48 hours after fortification with CMBF or HMBF. MFG size in each sample of fortified milk was measured by laser light scattering. MFG size in the fortified milks increased gradually over time. At 24 and 48 hours after fortification, MFG size in the milk with CMBF was larger than that in the milk with HMBF ($4.8 \pm 0.5$ vs $4.3 \pm 0.3$ μm, $p<0.01$, $5.1 \pm 0.7$ vs $4.5 \pm 0.4$ μm, $p = 0.03$, respectively). HMBF is associated with less alteration of MFG size than CMBF. This may have an impact on feeding tolerance of very preterm infants.

## Introduction

Mother's own milk (MOM) is the best nutrition for very preterm infants. There are many advantages, such as decreased rates of infection, necrotizing enterocolitis (NEC), retinopathy of prematurity, and better cognitive development [1]. However, the amount of protein, calcium, and phosphorus in MOM is inadequate for the appropriate growth of preterm infants. Therefore, a human milk fortifier (HMF) is necessary.

In Japan, the only HMF available for preterm infants is derived from cow's milk and is in a powder form, known as cow milk-based fortifier (CMBF, Morinaga Co. Ltd., Tokyo, Japan). Although fortification of human milk with CMBF may result in variable degrees of gastrointestinal (GI) symptoms, CMBF in powder form is widely used all over the world. One common, severe GI symptom is the formation of fatty acid calcium stones. Case reports on intestinal

**Funding:** KH receives speaker honoraria from Prolacta Bioscience. Prolacta Bioscience (www. prolacta.com) provided the product for use in this study but was not involved in the study design, data collection or analysis, performance of the study, outcomes, decision to publish, or manuscript writing.

**Competing interests:** Prolacta Bioscience (www. prolacta.com) provided the product for use in this study but was not involved in the study design, data collection or analysis, performance of the study, outcomes, decision to publish, or manuscript writing. There are no declarations regarding employment, consultancy, patents, products in development, or marketed products. This funding does not alter our adherence to all PLOS ONE policies on sharing data and materials.

obstruction by fatty acid calcium stones have been reported [2–4]. All of the reported infants with calcium stone obstruction were fed with human milk fortified by a CMBF. Additionally, many neonatologists are concerned about the tolerance of a powdered HMF in very preterm infants with a previous history of necrotizing enterocolitis (NEC), other bowel surgery, or feeding intolerance.

Recently, we reported the first use in Japan of a human milk-based fortifier (HMBF, Prolacta Bioscience, California, USA) as a successful rescue strategy for very preterm infants with severe meconium ileus who had undergone surgeries and were demonstrating poor feeding tolerance and growth [5]. Those infants did not demonstrate any GI problems with HMBF; therefore, we assumed HMBF leads to better fat digestion for these very premature infants with a history of GI tract surgeries. When human milk is fortified according to the manufacturers' recommendations, the energy and protein level are similar, although there are some differences in fat, carbohydrate, and calcium content.

Some premature infants may not tolerate HMF well. One possible reason is an alteration of the milk fat globule (MFG) in both shape and size. The MFG in unfortified human milk maintains its original shape at the end of gastric digestion [6] but little is known about the shape of MFG in fortified human milk. The MFG structure is also altered by temperature changes and increases in size when human milk is frozen [7, 8]. While the freeze-thaw process typical in many hospitals may change the pH of human milk, this change in pH did not seem to impact MFG size [6]. Additionally, significantly larger fat and mineral aggregations have been found in the feces of infants fed fortified human milk compared to those with preterm formula [9]. Therefore, we speculate that the alteration of the microstructure of the MFG of donor human milk during the freezing process could make aggregates with calcium when the HMF is added and be related to poorer digestion ability.

A safe strategy for the addition of HMF to thawed milk for very preterm infants with GI surgeries or demonstrating feeding intolerance should be established. Our objective was to find out if the HMBF affects MFG size less than CMBF, which may impact overall feeding tolerance.

## Methods

Unpasteurized donor human milk (DHM) was evaluated in vitro. All mothers who donated milk provided written, informed consent to use their milk for research purposes. The protocol was approved by the Institutional Review Board of the Showa University School of Medicine.

We collected a total of 60 mL frozen DHM from each mother. The DHM was thawed completely in the refrigerator for 20 hrs. Five mL of thawed samples were used for controls as unfortified milk. The rest of the milk was divided into aliquots with the addition of either 1 packet of CMBF to 30mL DHM or 6mL of HMBF added to 24mL DHM and mixed according to manufacturers' recommendation. The contents of the CMBF and HMBF along with 100 mL comparisons of fortified human milk are shown in Table 1.

MFG size measurements were taken at four times: baseline immediately upon thawing before fortification, and 1, 24, and 48 hours after fortification (Fig 1). Samples were kept in the refrigerator (4°C) until the measurements were performed. The MFG size was measured three times for each sample. The two variables that were evaluated included: 1) MV: Mean diameter of the "volume distribution" which represents the center of gravity of the distribution; and 2) MA: Mean diameter of the "area distribution" which is calculated from the volume distribution.

The measurement of MFG size by laser light scattering has been described elsewhere [7, 10–12]. In brief, Microtrac S3500 (Nikkiso, Tokyo, Japan) employs the laser light scattering

**Table 1. Contents in unfortified/fortified HM.**

|  | energy(Cal) | protein(g) | fat(g) | carbohydrate(g) | calcium(mg) |
|---|---|---|---|---|---|
| (　/content) |  |  |  |  |  |
| CMBF(/sachet) | 20 | 1 | 1 | 1.7 | 100 |
| HMBF(/30ml) | 43.1 | 1.8 | 2.8 | 2.7 | 103.1 |
| (　　/100mL) |  |  |  |  |  |
| CMBF(#1) | 89 | 2.3 | 4.7 | 9.5 | 132 |
| HMBF(#2) | 90.7 | 2.4 | 5.3 | 8.3 | 119.2 |

[#1]: 100ml HM + 1 sachet of CMBF

[#2]: 70 ml HM + 30 ml HMBF.

method for particle size measurement and covers from 0.02 to 2800 μM. The Tri-Laser system allows light scattering measurements to be made from the forward low angle region to almost the entire angular spectrum (approximately zero to 160 degree). It achieves this through a combination of three lasers and two detector arrays, all in fixed positions. The analysis of scattered light to determine particle size employs a Mie-based unified angular scattering theory from large particle analysis to small particle analysis.

An electron microscope evaluation was performed to observe the difference in MFG size between human milk samples. In terms of scanning electron microscopy, we prepared three samples: thawed unfortified human milk, human milk fortified with CMBF for 24 hours, and human milk fortified with HMBF for 24 hours. The samples were dropped onto a nano-percolator (JEOL, Tokyo, Japan) for 1 minute. The samples were air-dried and observed without metal coating by LVSEM (SU-1000; Hitachi High-Tech Co. Tokyo, Japan) using a backscattered electron mode with acceleration voltage of 15 kV in 30 Pa.

The data were summarized by means and standard deviations at each time point of evaluation (after freeze-thaw, 1, 24, and 48 hours post-fortification). To assess whether there were differences in the size distribution of MFG (dependent variable) over time between the two fortified groups (CMBF and HMBF; the independent variable), we used a two-way repeated measures analysis of variance model with an assumption of circularity for the variance-

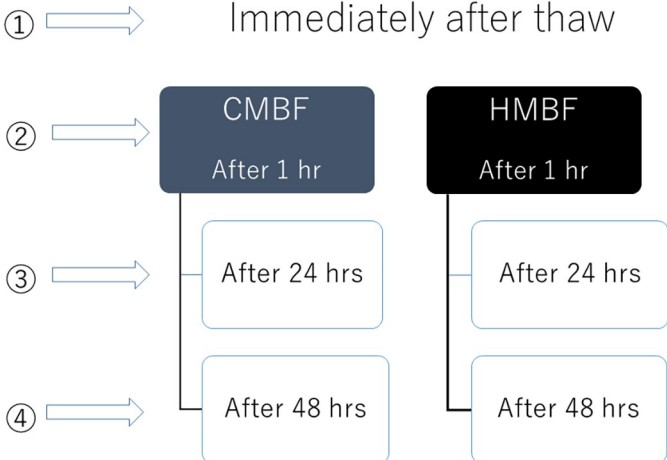

**Fig 1. The timing of particle size measurement.** Timing of Particle Size Measurement. MFG size measurements were taken at four times: baseline upon thawing before fortification, and 1, 24, and 48 hours after fortification.

covariance matrix of the data. The assumptions of normality and circularity/sphericity were checked by the Kolmogorov-Smirnov test and Machly's test, respectively. The lack of the latter assumption was adjusted using the Greenhouse-Geisser correction, if necessary. While the statistical model employed the main effects of fortifier group and time, the group-by-time interaction was the key term in this analysis.

Given the nature of the data, missing values were not an issue. Outliers were detected through the use of box plots for each fortifier group (where outliers were defined as values below the 25th percentile minus 1.5 times the interquartile range and above the 75th percentile plus 1.5 times the interquartile range). Since no outliers were detected, no data transformations or adjustment for outliers were required.

A p-value <0.05 was considered to be statistically significant for all inferential analyses. All statistical analyses were performed using NCSS 2021 Statistical Software (2021, NCSS, LLC. Kaysville, Utah, USA).

## Results

Samples of human milk were obtained from 9 Japanese women who were 27–36 years of age and 3–6 months postpartum. All the mothers were registered donors of the Japanese Human Milk Bank Association, had delivered term infants, and had exclusively breastfed their infants.

The mean MFG size in the thawed unfortified DHM and fortified DHM with either CMBF or HMBF is shown in Table 2. The MFG size distribution of the thawed, unfortified milk is shown in Fig 2. At baseline, mean particle size (MA) of MFG in the thawed, unfortified milk was 3.9 ± 0.4 μm. Fortification of 1 hour did not alter the MFG particle size significantly (4.1 ± 0.7 and 4.2 ± 0.2 μm, CMBF and HMBF, respectively). However, the MFG size became larger over time in both fortified milks. At 24 hours after fortification, MFG particle size in the milk with CMBF was larger than MFG size in the milk with HMBF (4.8 ± 0.5 vs 4.3 ± 0.3 μm, p<0.01). At 48 hours after fortification, MFG size remained larger in the milk with CMBF compared to the milk with HMBF (5.1 ± 0.7 vs 4.5 ± 0.4 μm, p = 0.03). In fact, MFG size of the milk with HMBF at 48 hours (4.5 ± 0.4 μm) was still smaller than that found in the milk with CMBF at 24 hours (4.8 ± 0.5 μm). S1 and S2 Tables show all data with HMBF and CMBF, respectively.

As shown in Fig 3, particle size increased over time from baseline for each of the fortified milks. The MV from baseline to 48 hours of the milk fortified with CMBF was significantly larger than that of the milk fortified with HMBF (p<0.001). Additionally, the MA from baseline to 48 hours of the milk fortified with CMBF was significantly larger than that of the milk

**Table 2. Changes in area average diameter (MA), volume average diameter (MV), of the MFG with time after fortification.** (n = 9).

|  | variables |  | 1 hour | 24 hours | 48 hours |
|---|---|---|---|---|---|
| Freeze-thawed (control) | MA | 3.9±0.4 |  |  |  |
|  | MV | 7.1±2.5 |  |  |  |
| CMBF | MA |  | 4.1±0.7 | 4.8±0.5[##] | 5.1±0.7[#] |
|  | MV |  | 11.1±4.1 | 16.8±5.4[##] | 18.5±5.1[###] |
| HMBF | MA |  | 4.2±0.2 | 4.3±0.3 | 4.5±0.4 |
|  | MV |  | 9.7±1.6 | 9.9±2.2 | 10.8±2.7 |

[#]: p<0.05,

[##]:p<0.01,

[###]: p<0.001 vs HMBF

MA (p<0.001) and MV (p = 0,002) increased with time course.

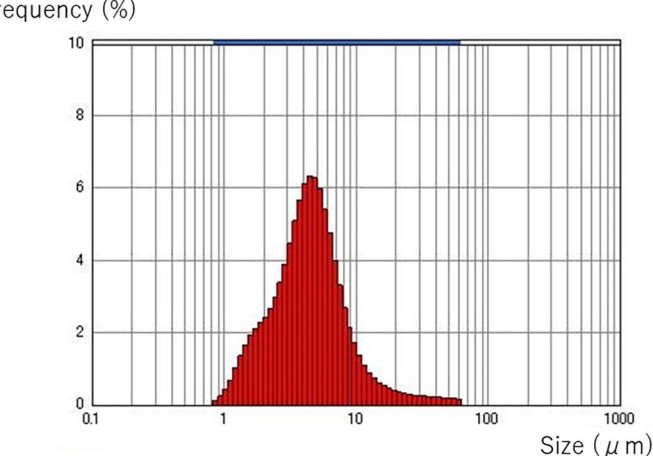

**Fig 2. MFG size distribution (thawed milk prior to fortification).** The particle size distribution after the freeze-thaw process prior to any fortification is shown.

fortified with HMBF (p = 0.002). The distribution of particle size as measured by electron microscopy at 24 hours after fortification for both CMBF and HMBF are shown in Fig 4. The shape of the particle distribution after 24 hours of CMBF fortification is different from the shape after 24 hours of HMBF fortification, with the CMBF fortification showing 2 peaks and a longer tail (Fig 5).

## Discussion

The feeding of very preterm infants with an exclusive human milk diet which uses only HMBF with DHM or MOM has been shown to reduce the incidence of NEC, sepsis, bronchopulmonary dysplasia, and retinopathy of prematurity compared to cow milk-based nutrition sources [13–16]. The results of the current study provide new data that the addition of HMBF to thawed DHM results in less alteration in MFG size compared to CMBF. Even after 48 hours of fortification, the MFG size in HM with HMBF was smaller than that in HM with CMBF at 24 hours after fortification.

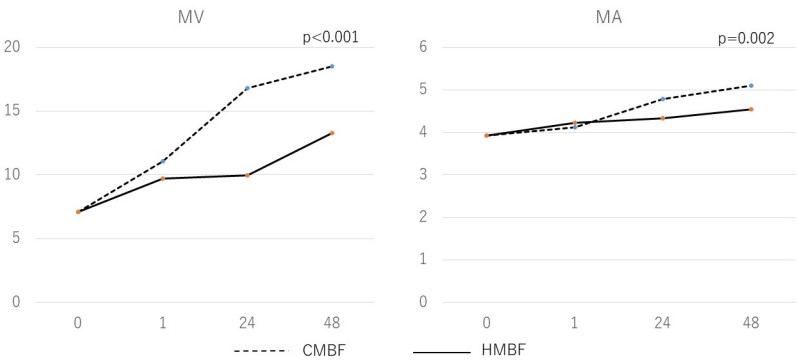

**Fig 3. Particle size with time course.** The diameter of MFG particles increased over time in donor human milk fortified with either cow milk-based fortifier (CMBF) or human milk-based fortifier (HMBF), with larger MFG particle sizes found in milk with CMBF. There were significant differences over time from baseline to 48 hours in mean diameter of the volume distribution (MV) and mean diameter of the area distribution (MA) (p<0.001 and p = 0.002, respectively) between CMBF and HMBF.

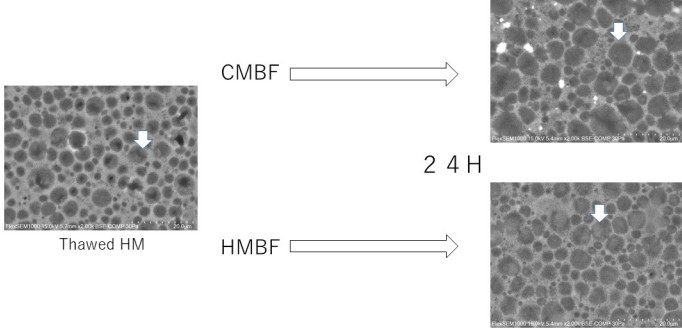

**Fig 4. MFG with electron microscope.** The electron microscopes for unfortified human milk (left side) and the fortified human milk at 24 hours after fortification are shown (right side; the upper photo indicates CMBF and the lower photo indicates HMBF). The arrow indicates MFGs.

Our recent questionnaire survey of nurses in Japan found that fortification with CMBF often resulted in reports of abdominal distension and increase in gastric residuals [17]. Similar reports have also been published [18, 19]. Hair et al. has shown that use of CMBF in the diet of very preterm infants correlates with more days on parenteral nutrition and a longer time to achieve full feeds [13]. The source of HMF has been known to affect feeding tolerance of very preterm infants [20]. Feeding intolerance in very preterm infants is a concern since weight gain during NICU stay is known to relate to cognitive function in later life [21].

A possible mechanism for GI distress symptoms is the use of previously frozen, thawed human milk since the freezing process alters the MFG membrane [3, 7]. Previous reports of MFG particle size in fresh unfrozen milk were shown to be 4.0 μm, and increasing in size the longer the milk was frozen, up to 12 months, longer than the milk samples in our study were frozen [7]. This is similar to the particle size we found immediately upon thawing the milk samples. The damaged MFG membrane and premature release of triglycerides could deteriorate fat digestion and the lipid status in milk has been known to affect gut digestion in a piglet model [22]. Because very preterm infants already have difficulty with fat absorption [23], these changes in MFG could further impact the infant's digestive function. Wei et al [24] previously concluded that MFG with high electrokinetic potential are electrically stabilized while MFG with low electrokinetic potentials tend to coagulate or flocculate. Human MFG demonstrate low electrokinetic potential compared to other animals and may be of special benefit to the digestion and metabolism of human milk fat in infants [25]. Although we have not measured electrokinetic

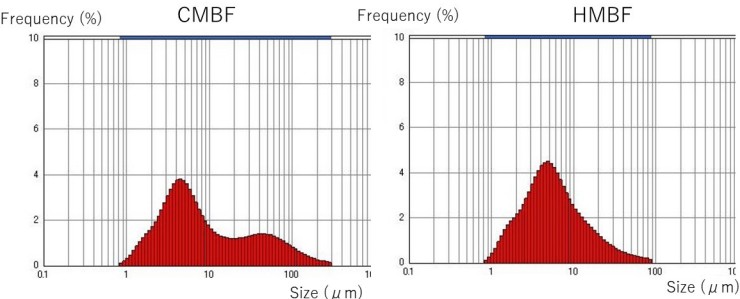

**Fig 5. Distribution of particle size: 24 hours after fortification.** The particle size distributions after 24 hours of fortification are shown for milk with cow milk-based fortifier (CMBF) and human milk-based fortifier (HMBF). The shape of the particle distribution after 24 hours of CMBF fortification was wider toward the end and different from the shape after 24 hours of HMBF fortification.

potential in this study, casein micelles protrude outside the MFG membrane and are likely to affect electrokinetic potential [25]. Casein micelles attach to the damaged MFG membrane resulting in larger MFG particle sizes. Since interfacial and electrostatic properties are being altered, damages to the MFG can profoundly affect the stability and properties of milk [25].

In this study, addition of CMBF to DHM resulted in larger MFG size compared with HMBF. This result could indicate that denatured MFG membranes make aggregates with calcium when CMBF is added to thawed human milk. Jensen et al. reported that significantly larger fat and mineral aggregations have been found in the feces of infants fed fortified human milk compared to those with preterm formula [9]. Our results show that the MFG size enlarged with time after fortification; therefore, the aggregation process could be exaggerated by the lag time from addition of HMF. Many NICUs in Japan add powdered CMBF to thawed frozen milk once a day for convenience, which leads to larger MFG size as shown in our results. Furthermore, CMBF contains beta-lactoglobulin which is not present in HMBF, and could also play a role in MFG alteration [26]. Although we cannot estimate how the size changes shown in our results have clinical significance, the more efficient gastric digestion of the fat in human milk might be related to the structure and size of the MFG.

Another issue of concern with feeding intolerance is osmolality which increases with time after the mixture of HMF with human milk [27] at the same time that there is a decrease in anti-infective factors in human milk [28]. For CMBF, osmolality increases to 360 mOsm/kgH2O immediately after fortification and then further increases to 400 mOsm/kgH2O after 24 hours. The osmolalities of human milk fortified with HMBF are in the range of 350–380 mOsm/KgH$_2$0, independent of time. Therefore, differences in osmolality does not appear to be significant between these two fortification strategies.

There are some limitations in this study. One limitation is that we did not evaluate a liquid CMBF because there is not yet one available in Japan as there is in other countries. Human milk fortified with liquid CMBF significantly improves preterm infant docosahexanoic acid and arachidonic acid status compared to powdered HMF [29] and may have a different outcome on alteration of the MFG size. Additionally, more data is needed to determine the mechanism of action whereby HMF changes the MFG particle size, the significance of the MFG particle size distribution, and its impact on feeding intolerance of the infant. Finally, we did not evaluate the difference in fat absorption between HMBF and CMBF in clinical settings. However, HMBF has been known to result in better GI tolerance compared to CMBF [20, 30]; therefore, we assume the fat absorption might be better tolerated with HMBF due to decreased MFG size alteration.

In conclusion, a HMBF is associated with less alteration of MFG size than CMBF. This may impact feeding tolerance of very preterm infants who often have difficulties with fat absorption and should be considered when providing nutritional care.

## Supporting information

**S1 Table. All data on MFG size with HMBF.**
(XLSX)

**S2 Table. All data on MFG size with CMBF.** MFG particle size data are shown 1 hour after fortification in upper panel, 24 hours after fortification in middle panel, and 48 hours after fortification in the bottom panel. MV: mean volume diameter, MN: mean number diameter, MA: mean area diameter, SD: standard deviation, CS: calculated specific surface area, TR: transmission.
(XLSX)

## Acknowledgments

The authors are grateful to the Japanese mothers and the Japanese Human Milk Bank Association for contributing human milk samples for the study.

## Author Contributions

**Conceptualization:** Yurika Yoshida, Minami Azuma, Haruhiro Kuwabara, Tokuo Miyazawa, Yuya Nakano, Kazuna Furukawa, Masahiko Izumizaki, Katsumi Mizuno.

**Data curation:** Yurika Yoshida, Minami Azuma, Haruhiro Kuwabara, Tokuo Miyazawa, Yuya Nakano, Kazuna Furukawa, Masahiko Izumizaki, Katsumi Mizuno.

**Formal analysis:** Yurika Yoshida, Minami Azuma, Haruhiro Kuwabara, Tokuo Miyazawa, Yuya Nakano, Kazuna Furukawa, Masahiko Izumizaki, Katsumi Mizuno.

**Funding acquisition:** Yurika Yoshida, Katsumi Mizuno.

**Investigation:** Yurika Yoshida, Minami Azuma, Haruhiro Kuwabara, Tokuo Miyazawa, Yuya Nakano, Kazuna Furukawa, Masahiko Izumizaki, Takashi Takaki, Mari Sakaue, Katsumi Mizuno.

**Methodology:** Yurika Yoshida, Minami Azuma, Haruhiro Kuwabara, Tokuo Miyazawa, Yuya Nakano, Kazuna Furukawa, Masahiko Izumizaki, Takashi Takaki, Mari Sakaue, Katsumi Mizuno.

**Project administration:** Yurika Yoshida, Minami Azuma, Haruhiro Kuwabara, Tokuo Miyazawa, Yuya Nakano, Kazuna Furukawa, Masahiko Izumizaki, Katsumi Mizuno.

**Resources:** Yurika Yoshida, Minami Azuma, Haruhiro Kuwabara, Tokuo Miyazawa, Yuya Nakano, Kazuna Furukawa, Masahiko Izumizaki, Katsumi Mizuno.

**Software:** Yuya Nakano, Katsumi Mizuno.

**Supervision:** Yurika Yoshida, Minami Azuma, Haruhiro Kuwabara, Tokuo Miyazawa, Kazuna Furukawa, Masahiko Izumizaki, Mari Sakaue, Katsumi Mizuno.

**Validation:** Yurika Yoshida, Minami Azuma, Haruhiro Kuwabara, Tokuo Miyazawa, Yuya Nakano, Kazuna Furukawa, Masahiko Izumizaki, Katsumi Mizuno.

**Visualization:** Yurika Yoshida, Minami Azuma, Haruhiro Kuwabara, Tokuo Miyazawa, Yuya Nakano, Kazuna Furukawa, Masahiko Izumizaki, Katsumi Mizuno.

**Writing – original draft:** Yurika Yoshida, Minami Azuma, Haruhiro Kuwabara, Tokuo Miyazawa, Yuya Nakano, Kazuna Furukawa, Keli M. Hawthorne, Masahiko Izumizaki, Takashi Takaki, Mari Sakaue, Katsumi Mizuno.

**Writing – review & editing:** Yurika Yoshida, Minami Azuma, Haruhiro Kuwabara, Tokuo Miyazawa, Yuya Nakano, Kazuna Furukawa, Keli M. Hawthorne, Masahiko Izumizaki, Takashi Takaki, Mari Sakaue, Katsumi Mizuno.

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
