## [Decision Letter · Decision Letter 0]

26 Mar 2021

PONE-D-21-01879

Human Milk-Based Fortifier is Associated with Less Alteration of Milk Fat Globule Size than Cow Milk-Based Fortifier

PLOS ONE

Dear Dr. Mizuno,

Thank you for submitting your manuscript to PLOS ONE. After careful consideration, we feel that it has merit but does not fully meet PLOS ONE’s publication criteria as it currently stands. Therefore, we invite you to submit a revised version of the manuscript that addresses the points raised during the review process.

We look forward to receiving your revised manuscript.

Kind regards,

Juan J Loor

Academic Editor

PLOS ONE

Journal Requirements:

2. To comply with PLOS ONE submission guidelines, in your Methods section, please provide additional information regarding your statistical analyses. For more information on PLOS ONE's expectations for statistical reporting, please see https://journals.plos.org/plosone/s/submission-guidelines.#loc-statistical-reporting.

4. Thank you for stating the following in the Financial Disclosure section:

'KH receives speaker honoraria from Prolacta Bioscience.

Prolacta Bioscience (www.prolacta.com) provided the product for use in this study but

was not involved in the study design, data collection or analysis, performance of the

study, outcomes, decision to publish, or manuscript writing.'

We note that you received funding from a commercial source: Prolacta Bioscience

Within this Competing Interests Statement, please confirm that this does not alter your adherence to all PLOS ONE policies on sharing data and materials by including the following statement: "This does not alter our adherence to PLOS ONE policies on sharing data and materials.” (as detailed online in our guide for authors http://journals.plos.org/plosone/s/competing-interests).  If there are restrictions on sharing of data and/or materials, please state these. Please note that we cannot proceed with consideration of your article until this information has been declared

5. Please upload a copy of Figures 1 to 4, to which you refer in your text. If the figures are no longer to be included as part of the submission please remove all reference to them within the text.

Additional Editor Comments (if provided):

Reviewers' comments:

Reviewer's Responses to Questions

**Comments to the Author**

1. Is the manuscript technically sound, and do the data support the conclusions?

Reviewer #1: Partly

Reviewer #2: Partly

2. Has the statistical analysis been performed appropriately and rigorously? 

Reviewer #1: Yes

Reviewer #2: Yes

3. Have the authors made all data underlying the findings in their manuscript fully available?

Reviewer #1: Yes

Reviewer #2: No

4. Is the manuscript presented in an intelligible fashion and written in standard English?

Reviewer #1: Yes

Reviewer #2: Yes

5. Review Comments to the Author

Reviewer #1: This study aimed to assess the change of the MFG size with the fortification of a human milk-based or a cow milk-based fortifier. Changes in the particle size might associate with feeding intolerance. This is of great relevance, particularly for low-birth-weight or preterm infants.

Over the manuscript, there are some inconsistencies in the grammar, as well as some repetitiveness in some parts. I would suggest authors revise the manuscript.

The Introduction section seems very broad. In this part, the authors should describe more about the relevance of the MFG size, the differences in composition of the two fortifiers, etc. This would create a greater interest in the reader.

Regarding the methodology, was the MGF size measurement done in duplicate, or were they single measurements? What software was used to conduct the statistical analyses?

Could the author summarize the demographic characteristics of the donors (e.g. age, weight, BMI)?

The results section mentions three different figures which were not part of the manuscript. I wonder if there was an issue with the website/system. I can not discuss much about the information associated with the figures.

Table 1 could include more information to make it more comprehensive. For example, could you add the number of samples in each group (n=), as well as the p values for the differences within and between groups?

The discussion summarizes the findings and the impact on MFG sizes. Although the authors discuss some of the possible mechanisms, they could further explain how human milk-based or a cow milk-based fortifier changes the size and structure of the MFG and how the consumption affects the feeding tolerance in infants consuming these fortifiers.

Reviewer #2: Review Comments to the Author

This paper aimed at to evaluate the HMBF and CMBF on the MFG, and to explain it impact on feeding tolerance. The result sounds interesting, however, the paper still explain the detail of the test and how it works, more data need before it could accept. Question is list below:

1. For the particle size of fat globule, many parameter many effect the result, such as p

H, Ca and casein content in milk. You should add this data in the paper.

2. Line 76-80, you collected human milk from 9 monther, do you mixed the 9 sample together or

Separated? If you separate, does all sample the protein is 2.9g/100ml? for the line 79, more information about CMBF is need, company, composition.

3. Line 82-83, after you mixed together, you test the size in 0, 1, 24, 48h. Does the sample is put in refrigerator? What’s temperature you keep the sample?

4. Excepted the particle size, such as the scanning electron microscope is need to show the difference of the samples.

5. The figures is not show in the paper, please added it.

6. PLOS authors have the option to publish the peer review history of their article (what does this mean?). If published, this will include your full peer review and any attached files.

Reviewer #1: No

Reviewer #2: **Yes: **Daxi Ren

---

## [Author Response · Author response to Decision Letter 0]

29 Jun 2021

Journal Requirements:

Updated to match requirements

2. To comply with PLOS ONE submission guidelines, in your Methods section, please provide additional information regarding your statistical analyses. For more information on PLOS ONE's expectations for statistical reporting, please see https://journals.plos.org/plosone/s/submission-guidelines.#loc-statistical-reporting.

Added to the manuscript.

We obtained written, informed consent from all mothers to use their milk for research purposes (added to the manuscript). We did not include any minors. 

We did not include a retrospective chart review or utilized archived samples as part of this study. 

4. Thank you for stating the following in the Financial Disclosure section:

'KH receives speaker honoraria from Prolacta Bioscience.

Prolacta Bioscience (www.prolacta.com) provided the product for use in this study but was not involved in the study design, data collection or analysis, performance of the study, outcomes, decision to publish, or manuscript writing.'

We note that you received funding from a commercial source: Prolacta Bioscience

Within this Competing Interests Statement, please confirm that this does not alter your adherence to all PLOS ONE policies on sharing data and materials by including the following statement: "This does not alter our adherence to PLOS ONE policies on sharing data and materials.” (as detailed online in our guide for authors http://journals.plos.org/plosone/s/competing-interests). If there are restrictions on sharing of data and/or materials, please state these. Please note that we cannot proceed with consideration of your article until this information has been declared

We have included the appropriate Competing Interests Statement and information within our updated cover letter. 

5. Please upload a copy of Figures 1 to 4, to which you refer in your text. If the figures are no longer to be included as part of the submission please remove all reference to them within the text.

Thank you, Figures 1-4 will be uploaded. 

. Review Comments to the Author

Reviewer #1: This study aimed to assess the change of the MFG size with the fortification of a human milk-based or a cow milk-based fortifier. Changes in the particle size might associate with feeding intolerance. This is of great relevance, particularly for low-birth-weight or preterm infants.

Over the manuscript, there are some inconsistencies in the grammar, as well as some repetitiveness in some parts. I would suggest authors revise the manuscript.

A native English speaker has provided writing and editing assistance for the manuscript to eliminate errors and repetitiveness. 

The Introduction section seems very broad. In this part, the authors should describe more about the relevance of the MFG size, the differences in composition of the two fortifiers, etc. This would create a greater interest in the reader.

More information about the MFG size has been added to the introduction. A comparison of the 2 fortifiers has been added in Table 1. 

Regarding the methodology, was the MGF size measurement done in duplicate, or were they single measurements? 

The MFG size was measured three times for each sample.

What software was used to conduct the statistical analyses?

All statistical analyses were performed using NCSS 2021 Statistical Software (2021, NCSS, LLC. Kaysville, Utah, USA).

Could the author summarize the demographic characteristics of the donors (e.g. age, weight, BMI)?

Samples of human milk were obtained from 9 Japanese women who were 27-36 years of age and 3-6 months postpartum. This has been added to the Results section. 

We did not obtain other demographic information such as weight or BMI. 

The results section mentions three different figures which were not part of the manuscript. I wonder if there was an issue with the website/system. I can not discuss much about the information associated with the figures.

We are sorry for not having attached the figures. In revised manuscript, we confirmed that we could upload the figures appropriately.

Table 1 could include more information to make it more comprehensive. For example, could you add the number of samples in each group (n=), as well as the p values for the differences within and between groups?

We added the number of samples in each group as well as the p values for the differences in Table 2.

The discussion summarizes the findings and the impact on MFG sizes. Although the authors discuss some of the possible mechanisms, they could further explain how human milk-based or a cow milk-based fortifier changes the size and structure of the MFG and how the consumption affects the feeding tolerance in infants consuming these fortifiers.

The alteration of the microstructure of MFG with cold temperatures and with a cow milk-based fortifier in part relates to poorer digestion ability. We are uncertain of the mechanism causing this effect. More data is needed to determine this mechanism of action. 

Reviewer #2: Review Comments to the Author

This paper aimed at to evaluate the HMBF and CMBF on the MFG, and to explain it impact on feeding tolerance. The result sounds interesting, however, the paper still explain the detail of the test and how it works, more data need before it could accept. Question is list below:

1. For the particle size of fat globule, many parameter many effect the result, such as pH, Ca and casein content in milk. You should add this data in the paper.

Thank you for this helpful comment. We have added the data on pH, calcium, osmolarity and casein content in fortified milk for both types of fortifiers to Table 1.

2. Line 76-80, you collected human milk from 9 mother, do you mixed the 9 sample together or Separated? If you separate, does all sample the protein is 2.9g/100ml? for the line 79, more information about CMBF is need, company, composition.

We did not mix the samples. We fortified the donor milk according to the manufactures’ recommendation. The protein content 2.9g/100ml was estimated from the average protein content of human milk; we deleted this part in the revision.

3. Line 82-83, after you mixed together, you test the size in 0, 1, 24, 48h. Does the sample is put in refrigerator? What’s temperature you keep the sample?

Yes, we kept the sample in a refrigerator at 4℃

4. Excepted the particle size, such as the scanning electron microscope is need to show the difference of the samples.

We performed the electron microscope experiment as the reviewer suggested and it shows the difference in MFG size visually. We appreciate this suggestion.

5. The figures is not show in the paper, please added it.

We apologize for not having attached the figures. Now attached.

---

## [Decision Letter · Decision Letter 1]

3 Sep 2021

Human Milk-Based Fortifier is Associated with Less Alteration of Milk Fat Globule Size than Cow Milk-Based Fortifier

PONE-D-21-01879R1

Dear Dr. Mizuno,

We’re pleased to inform you that your manuscript has been judged scientifically suitable for publication and will be formally accepted for publication once it meets all outstanding technical requirements.

Kind regards,

Juan J Loor

Academic Editor

PLOS ONE

Additional Editor Comments (optional):

Reviewers' comments:

Reviewer's Responses to Questions

**Comments to the Author**

1. If the authors have adequately addressed your comments raised in a previous round of review and you feel that this manuscript is now acceptable for publication, you may indicate that here to bypass the “Comments to the Author” section, enter your conflict of interest statement in the “Confidential to Editor” section, and submit your "Accept" recommendation.

Reviewer #1: All comments have been addressed

Reviewer #2: All comments have been addressed

2. Is the manuscript technically sound, and do the data support the conclusions?

Reviewer #1: Yes

Reviewer #2: Yes

3. Has the statistical analysis been performed appropriately and rigorously? 

Reviewer #1: Yes

Reviewer #2: Yes

4. Have the authors made all data underlying the findings in their manuscript fully available?

Reviewer #1: Yes

Reviewer #2: Yes

5. Is the manuscript presented in an intelligible fashion and written in standard English?

Reviewer #1: Yes

Reviewer #2: Yes

6. Review Comments to the Author

Reviewer #1: Thank you to the authors for addressing the comments and suggestions. The manuscript has improved.

There are only some minor things that I would suggest. The numbers for the citations should be before the point or commas just so there are no blank spaces. The title of Figure 3 should begin with a capital letter.

Reviewer #2: (No Response)

7. PLOS authors have the option to publish the peer review history of their article (what does this mean?). If published, this will include your full peer review and any attached files.

Reviewer #1: **Yes: **Miriam Aguilar-Lopez

Reviewer #2: No

---

## [Editor Report · Acceptance letter]

25 Nov 2021

PONE-D-21-01879R1 

Human milk-based fortifier is associated with less alteration of milk fat globule size than cow milk-based fortifier 

Dear Dr. Mizuno:

I'm pleased to inform you that your manuscript has been deemed suitable for publication in PLOS ONE. Congratulations! Your manuscript is now with our production department. 

Kind regards, 

on behalf of

Dr. Juan J Loor 

Academic Editor

PLOS ONE